# Identifying Parametric Models Used to Estimate Track Irregularities of a High-Speed Railway

Sunghoon Choi

High-Speed Railroad Research Department, Korea Railroad Research Institute, Uiwang-si 16105, Republic of Korea; schoi@krri.re.kr

**Abstract:** This study aims to identify parametric models to estimate track irregularities in high-speed railways with simple acceleration measurements. The primary contribution of current research is the development of effective parametric models with smaller parameters. These parameters are derived from the measured data via a specialized track geometry inspection system. An adaptive Kalman filter algorithm, using the displacement estimated from the acceleration signals as the input and measured track irregularities as the output, is applied to obtain the model's unknown parameters. These models are applied to acceleration measured from high-speed rail vehicles in operation, and track irregularities are estimated in spatial and wavelength domains. The estimated irregularities are compared to the track geometry inspection system's results.

**Keywords:** track irregularities; accelerometers; parametric models; system identification; adaptive Kalman filter; IIR model; FIR model; Hybrid model

## 1. Introduction

Track irregularities mainly originate from heavy traffic loads and unexpected ground movement, which excite the wheelset and create unwanted vibrations. These vibrations may affect passenger comfort and cause the degradation of track and train components or, at worst, derailment. Therefore, rail operators and infrastructure managers should regularly monitor and maintain the track irregularities with proper methods to ensure passenger comfort and safety.

Track irregularities, defined as deviations from the original track geometry, are usually measured with a track inspection vehicle equipped with special measurement devices using contact probes, lasers, or optical sensors [1,2]. However, using this inspection vehicle with measuring devices is costly and complicated. In addition, the dynamic characteristics of a track inspection vehicle are different from those of a high-speed train in commercial operation, so the dynamic deflection of the track caused by a high-speed train cannot be appropriately measured. As an alternative, a simple and inexpensive device using accelerometers installed on an in-service vehicle has drawn attention because it can be used for daily monitoring of track conditions. Theoretically, displacement can simply be estimated by double integrating acceleration. However, in practice, the integration usually yields unwanted drifts due to the non-zero initial condition of the signal, the direct current (DC) offsets, or the noise due to electrical/mechanical hysteresis in sensors or cables [3]. To avoid these drawbacks, many researchers have proposed model-based methods. In these methods, system models are required, which can represent the input–output relationship between acceleration and track irregularities.

Several studies have sought models that describe the dependence between acceleration and track irregularities. Kawasaki et al. [4] presented a method using car body acceleration and an auto-regression model with extra inputs. The properties of the suspension system highly influence the car body acceleration; hence, it is difficult to separate the effect of suspension from the acceleration signals. Weston et al. [5,6] used bogie-mounted

accelerometers and a gyroscope to monitor track irregularities and proposed a correction using the second-order dynamic model to improve the lateral irregularity. The results of such studies were limited to wavelengths less than 35 m or 70 m and thus were unsuitable for measuring long wavelengths up to 150 or 200 m. Alfi et al. [7] proposed a method for calculating irregularities with long wavelengths from vehicle acceleration measurements using a model-based identification procedure defined in the frequency domain. However, there was a non-negligible difference in the trend of the power spectral density and in the space diagram for wavelengths up to 120 m, which is essential for passenger comfort during high-speed journeys. Czop et al. [8] presented an approach to detect track irregularities using axle box accelerometers and the inverse linear parametric vehicle dynamics model. They focused on the relationship between the measured bending moment and axle box vibrations, not displacement, which is essential for making a track maintenance decision. Hidalgo et al. [9] and Tsunashima et al. [10] developed Kalman-filter-based techniques that combined a kinematic model and a dynamic model to identify track irregularities. Both works used an accelerometer and a gyroscope to estimate the vertical track irregularities, and relatively acceptable accuracy was obtained. Muñoz et al. [11] proposed an efficient Kalman-based methodology for monitoring lateral track irregularities using inertial sensors installed on a train in operation. In this study, two accelerometers are utilized to measure the lateral acceleration of the wheelset and the bogie frame. At the same time, a gyroscope is employed to detect the yaw angular velocity of the wheelset.

Lee et al. [12] proposed a mixed filtering approach using the Kalman, band-pass, and compensation filters for waveband monitoring of lateral and vertical track irregularities, which used accelerometers installed on the axle box and the bogie. The method used Kalman and band-pass filters for displacement estimation from measured acceleration. Compensation filters consisting of finite impulse response models with 40 parameters were used to correct for amplitude and phase difference, which result from the inherent characteristics of the preceding filters and the lateral motion of the wheelset or the bogie with respect to the track. However, the models were expensive and complex because too many parameters were used. This research proposes efficient parametric models with fewer parameters. The parameters are derived from the measured signals obtained using a track geometry inspection system (TGIS). An adaptive Kalman filter algorithm is applied to obtain the unknown parameters of track irregularities with an estimated displacement from acceleration signals as the input and track irregularity signals as the output. Finally, the developed models are used in the analysis of acceleration data measured from the axle box and the bogie of a high-speed train in operation.

This paper is organized as follows: The measurement setup used to obtain acceleration signals and track geometry is described in Section 2; Section 3 presents the process of estimating displacement from acceleration signals, while Section 4 explains the parametric models and the methodology used to estimate the parameters; Section 5 describes the model section process and validates the selected model; In Section 6, track irregularities are estimated using acceleration signals obtained from high-speed trains in operation, and the results are compared with the reference irregularities; and the summary and conclusions can be found in Section 7.

## 2. Measurement Setup

Figure 1 depicts the installation of lateral and vertical accelerometers on an axle box and a bogie of a high-speed train in operation. Capacitive-type accelerometers were used to measure low-frequency vibrations. The sampling frequency used to acquire each signal was 2048 Hz. Additionally, the train's speed was measured synchronously with the acceleration signals, so the filtered signals were rearranged from the time domain to the wavenumber domain with a 0.25 m sampling interval.

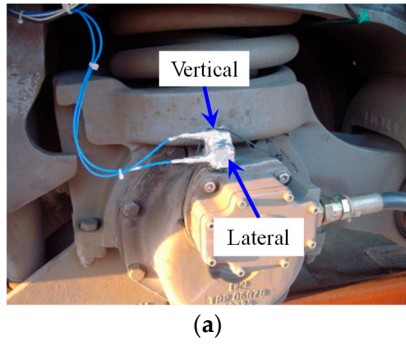 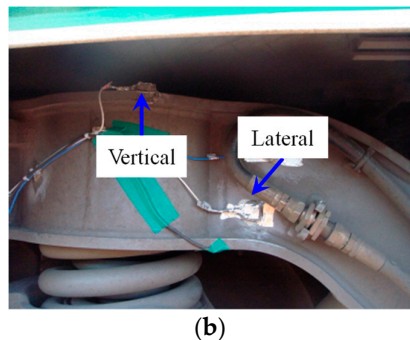

(**a**)                (**b**)

**Figure 1.** Accelerometers on axle box and bogie: (**a**) axle box-mounted accelerometers; (**b**) bogie-mounted accelerometers [12].

The profile and geometry of the track were measured using a specialized TGIS, which can operate at speeds of up to 320 km/h with a spatial resolution of 0.25 m. The TGIS was installed on the high-speed train HSR-350x, which is the prototype of commercial high-speed trains in Korea [13]. A light-sectioning technique using a small laser band as a sharp edge was utilized to measure the profile. The inertial data obtained with gyros and accelerometers, which were installed close to the laser device and the camera, were used for geometric measurements. Their signals underwent time-domain processing prior to conversion to the wavenumber domain through filtering and re-sampling.

## 3. Displacement Estimation from Acceleration

As mentioned in the introduction, displacement estimation from noisy acceleration using direct double integration results in unrealistic errors. A discrete state-space model and the Kalman filter were introduced to resolve the error in the previous works [12,14]. The following describes the state-space model for displacement estimation from noisy acceleration:

- State model

$$x_{n+1} = Fx_n + Gu_n, \tag{1}$$

- Space model

$$a_n = Hx_n + w_n, \tag{2}$$

where in

$$F = \begin{bmatrix} \alpha & 0 & 0 \\ 1 & 0 & 0 \\ 0 & 1 & 0 \end{bmatrix}, G = \begin{bmatrix} 1 \\ 0 \\ 0 \end{bmatrix}, \text{ and } H = \frac{1}{T_s^2} \begin{bmatrix} 1 & -2 & 1 \end{bmatrix}. \tag{3}$$

In Equation (3), $\alpha$ is a model parameter ($0 \ll \alpha \leq 1$) and $T_s$ is the sampling time. In the state-space equations, the state transition matrix $F$ is used to update the preceding state, and $G$ is the noise-input matrix. At the same time, $H$ is the measurement matrix used to map the estimated displacement onto the measured acceleration. The noises $u_n$ and $w_n$ are comprised of zero-mean white Gaussian processes. It is assumed that the initial displacement $x_0$ is zero.

The measured acceleration signals are utilized to estimate the displacement using a Kalman filter algorithm, and its covariance form is described as follows [15]:

- Initial condition

$$\hat{x}_{0|-1} = 0, \tag{4}$$

$$P_{0|-1} = \Pi_0. \tag{5}$$

- Recursion relations

- Innovations:

$$e_n = a_n - H\hat{x}_{n|n-1}, \tag{6}$$

- Innovation covariance:

$$R_{e,n} = HP_{n|n-1}H^* + R_n, \tag{7}$$

- Kalman prediction gain:

$$K_{p,n} = \left( FP_{n|n-1}H^* + GS_n \right) R_{e,n}^{-1}, \tag{8}$$

- State estimation:

$$\hat{x}_{n+1|n} = F\hat{x}_{n|n-1} + K_{p,n}e_n, \tag{9}$$

- State error covariance:

$$P_{n+1|n} = FP_{n|n-1}F^* + GQ_nG - K_{p,n}R_{e,n}K_{p,n}^*. \tag{10}$$

where $\hat{x}_{n+1|n}$ is the estimate of $x_n$, $P_{n+1|n}$ is the state error covariance information at step $n$, $\Pi_0$ is the auto-covariance of the initially estimated displacement $\hat{x}_{0|-1}$, and $Q_n$ and $R_n$ are the auto-covariances of $u_n$ and $w_n$, respectively.

After applying the Kalman filter, the third-order Butterworth band-pass filters are applied to eliminate the short-wavelength effect due to the wheel and the bogie and the long-wavelength effect due to the track's curves. Block diagrams illustrating the processes are presented in Figure 2.

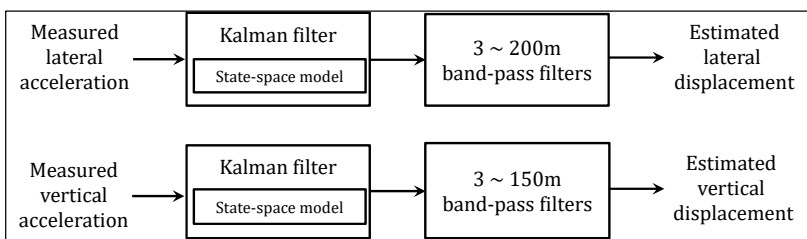

**Figure 2.** Displacement estimation from acceleration signals.

## 4. Model Setup and Identification

### 4.1. Concept

System identification is a technique used to build and complement a model with measurements [16]. The model may have a parametric or a non-parametric description in the frequency or time domains. The non-parametric description uses a system's impulse or frequency response with no parameters. It is highly dependent on time or frequency resolution. Using window techniques to reduce leakage can lead to signal distortion. Moreover, it is difficult to use in real-time calculation because an inverse Fourier transform is required to represent the results in the time domain. Therefore, a parametric representation was utilized in this work. It uses a model whose parameters are identified by an adaptive filtering algorithm that minimizes the error between the estimated and measured outputs. There are two kinds of parametric models: tailor-made and ready-made. The former is constructed using fundamental physical principles, and its parameters are unknown variables used to represent the characteristics of input–output relationships. The latter is built in terms of input, output, and transfer characteristics without any physical principles representing its internal workings. The variables describe the relationship between the inputs and outputs without representing physical quantities.

The relationship between track irregularities and the motion of an axle box or a bogie depends on the dynamic characteristics of the suspension system and the effective wheel conicity, the mechanism of which is highly complex; hence, it is challenging to represent using physical models [17]. Therefore, ready-made parametric models, which can predict track irregularities from the estimated displacement, are selected for this research. The displacement is estimated from the acceleration signals, which the TGIS measured so that they are synchronized with track irregularities. The estimated displacement was used instead of the measured acceleration to reduce uncertainty in the system identification.

### 4.2. Model Setup

The finite impulse response (FIR) and the infinite impulse response (IIR) models are considered for the ready-made parametric models. The FIR models depend on the present and previous values of input signals only, while the IIR models rely on one or more prior output values in addition to the input signal [18]. They are represented as follows:

- IIR model:

$$y_n = -\sum_{k=1}^{N} a_k y_{n-k} + \sum_{k=0}^{M} b_k x_{n-k}, \tag{11}$$

- FIR model:

$$y_n = \sum_{k=0}^{M} b_k x_{n-k}, \tag{12}$$

where $a_k$ and $b_k$ are model parameters, and $x_n$ and $y_n$ are the input and output of the model.

The FIR models have been widely used because of their stable behavior and good convergence in the estimation. However, they require many parameters and complex calculations to achieve a satisfactory performance. The IIR models can realize a sharp transition band with relatively few parameters, although they are unstable [19]. In this work, a hybrid model using a serial application of IIR or FIR models is applied to employ the advantages of both models. The parameters are initially set to zero and identified using the adaptive Kalman filter as illustrated in Figure 3a,b. In the figures, A and B correspond to IIR and FIR filters, respectively.

### 4.3. Adaptive Kalman Filter

Before applying an adaptive Kalman filter process, the parametric models are modified and described as a state-space model [20]:

- State model

$$\boldsymbol{\theta}_{n+1} = \boldsymbol{\theta}_n + \boldsymbol{u}_n, \tag{13}$$

- Space model:

$$y_n = \boldsymbol{H}_n^T \boldsymbol{\theta}_n + \boldsymbol{w}_n, \tag{14}$$

where

$$\boldsymbol{\theta}_n = [b_0, \ldots, b_M, a_1, \ldots, a_N]^T, \boldsymbol{H}_n = [x_n, \ldots x_{n-M}, -y_{n-1}, \ldots, -y_{n-N}]^T$$
$$\boldsymbol{u}_n, \boldsymbol{w}_n : Zero - mean \ white \ Gaussian \ noises.$$

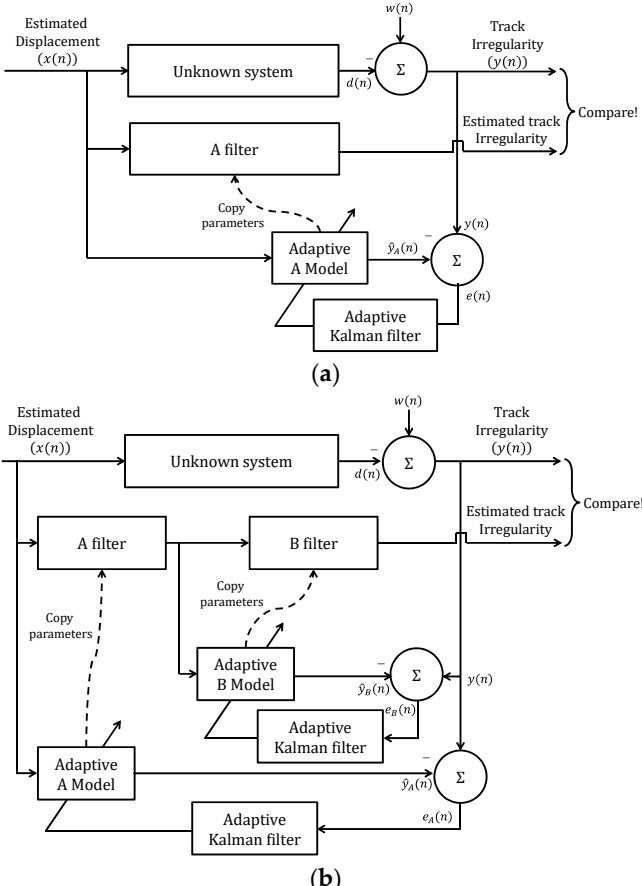

**Figure 3.** Adaptive filtering for system identification: (**a**) single model; (**b**) hybrid model.

The adaptive Kalman filter is applied to identify the model parameters recursively. It is an optimal estimation of the state that minimizes the squared error between the estimated and the measured track irregularities. The algorithm can be summarized as follows:

- Initialization

$$\hat{\boldsymbol{\theta}}_{0|-1} = 0, \boldsymbol{P}_{0|-1} = \boldsymbol{\Theta}_0, \tag{15}$$

- Recursions
- Innovations:

$$e_n = y_n - \boldsymbol{H}_n \hat{\boldsymbol{\theta}}_{n|n-1}, \tag{16}$$

- Innovation covariance:

$$\boldsymbol{R}_{e,n} = \boldsymbol{H}_n \boldsymbol{P}_{n|n-1} \boldsymbol{H}_n^* + \boldsymbol{R}_n, \tag{17}$$

- Kalman prediction gain:

$$\boldsymbol{K}_{p,n} = \boldsymbol{P}_{n|n-1} \boldsymbol{H}_n^* \boldsymbol{R}_{e,n}^{-1}, \tag{18}$$

- State estimation:

$$\hat{\boldsymbol{\theta}}_{n+1|n} = \hat{\boldsymbol{\theta}}_{n|n-1} + \boldsymbol{K}_{p,n} e_n, \tag{19}$$

- State error covariance:

$$P_{n+1|n} = P_{n|n-1} + Q_n - K_{p,n}R_{e,n}K_{p,n}^* = P_{n|n-1} + Q_n - K_{p,n}H_nP_{n|n-1}. \tag{20}$$

In the algorithm, $\hat{\theta}_{n+1|n}$ is the parameter estimation at $n$, and $Q_n$, $R_n$, and $\Theta_0$ represent the auto-covariance of $u_n$, $w_n$, and $\theta_0$, respectively.

## 5. Identification of the Models

### 5.1. Model Selection

To determine the model parameters, the displacements and irregularities obtained over a track section of 30 km in length, as shown in Figure 4, are used as the input and output. They were sampled every 0.25 m; hence, 120,000 points are used for identification in each direction. Several single and hybrid models are created using a selection of poles and zeros. The Pearson's product-moment correlation coefficient (PPMCC) between the estimated and the measured track irregularities, the measure of the linear dependency between the two signals [21], is evaluated to determine the optimal orders of the model. The mean square error (MSE) of the estimated irregularity is also evaluated to compare the performance of the models.

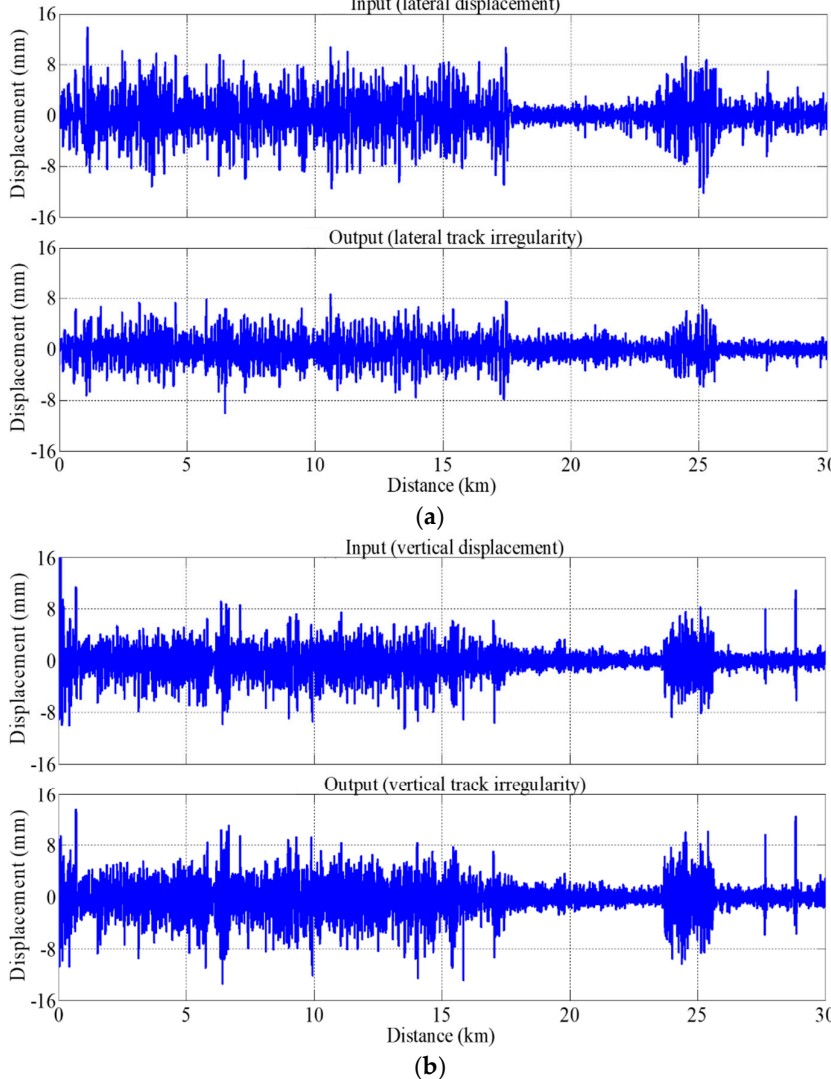

**Figure 4.** Input (displacement) and output (irregularity) signals for system identification: (**a**) lateral direction; (**b**) vertical direction.

The number of zeros ($M$) and poles ($N$) for the models, defined as in Equations (11) and (12), range from 0 to 20 and 1 to 20, respectively. The contour plots, as shown in Figure 5, are used for the intuitive evaluation of the models. The highest values are obtained above six zeros and one pole in the lateral direction. In the vertical direction, they are obtained above fourteen zeros and one pole, above twelve zeros and three poles, and above six zeros and five poles. IIR models with six zeros and one pole and six zeros and five poles are selected for the vertical and lateral directions, respectively, because models with smaller orders are efficient.

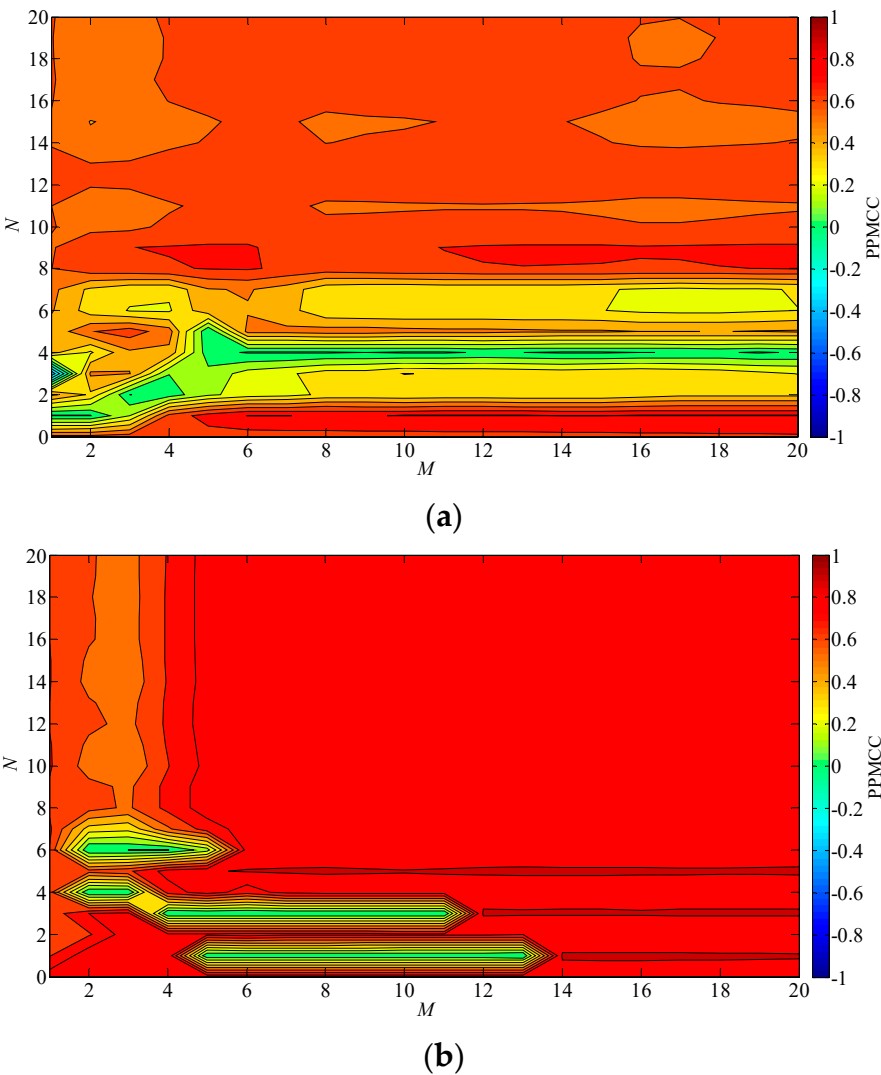

**Figure 5.** PPMCC contour plots using estimated displacement from measured acceleration: (**a**) lateral direction; (**b**) vertical direction.

The PPMCC and the MSE for different models are compared, as shown in Figure 6. The numbers in the parentheses indicate the number of zeros and poles; for example, FIR(4) represents the FIR model with $M = 4$, and IIR(6,5) stands for the IIR model with $M = 6$ and $N = 5$. The FIR(40) model is selected for comparison, as it was used in a previous work [11]. The FIR(4) model is also selected because it has the lowest number of zeros with no pole and a relatively high PPMCC in the lateral direction.

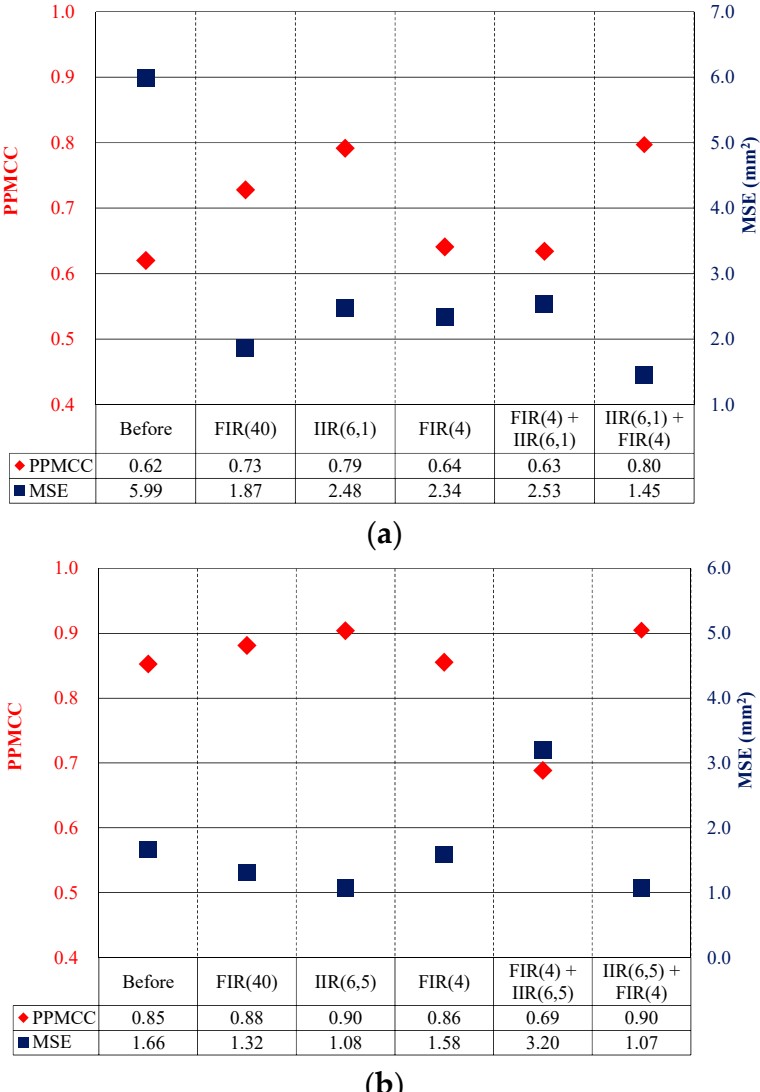

**Figure 6.** Comparison of PPMCC (◆) and MSE (■) of different models: (**a**) lateral direction; (**b**) vertical direction.

In Figure 6a, for the lateral direction, it is found that the PPMCC value is increased when a single IIR(6,1) or FIR(4) is used. However, the MSE for the IIR(6,1) model is higher than that of the FIR(4) or FIR(40) models. A hybrid IIR/FIR model, which applies the FIR(4) after the IIR(6,1), results in the highest PPMCC and the lowest MSE. On the contrary, the worst result is obtained when the sequence of the filters is reversed, i.e., the IIR(6,1) is applied after the FIR(4). Hence, it is confirmed that the hybrid IIR/FIR model is adequate and appropriate for estimating lateral track irregularity. In Figure 6b, for the vertical direction, it is found that the PPMCC and the MSE slightly improve when a single IIR(6,5) model is used. The effect of using a hybrid IIR/FIR model is moderate, and a hybrid FIR/IIR model worsens the result. Therefore, it is confirmed that a single IIR model would be sufficient to estimate the vertical track irregularity.

The spatial frequency responses of the derived models are shown in Figure 7. As mentioned in the introduction, the models are used to compensate for the discrepancies caused by the lateral motion of the axle box or the bogie relative to the track and the phase delay of the previous filters. Therefore, the model in the lateral direction has larger value than that in the vertical direction. Data from two additional test campaigns, carried out within a six-month interval after the first test, are used to ensure the reproducibility of the proposed models. Despite the six-month interval between the first and the third trials, the

frequency responses of both directions are consistent. Therefore, it is safe to conclude that the models are reliable and can be applied to estimate track irregularities.

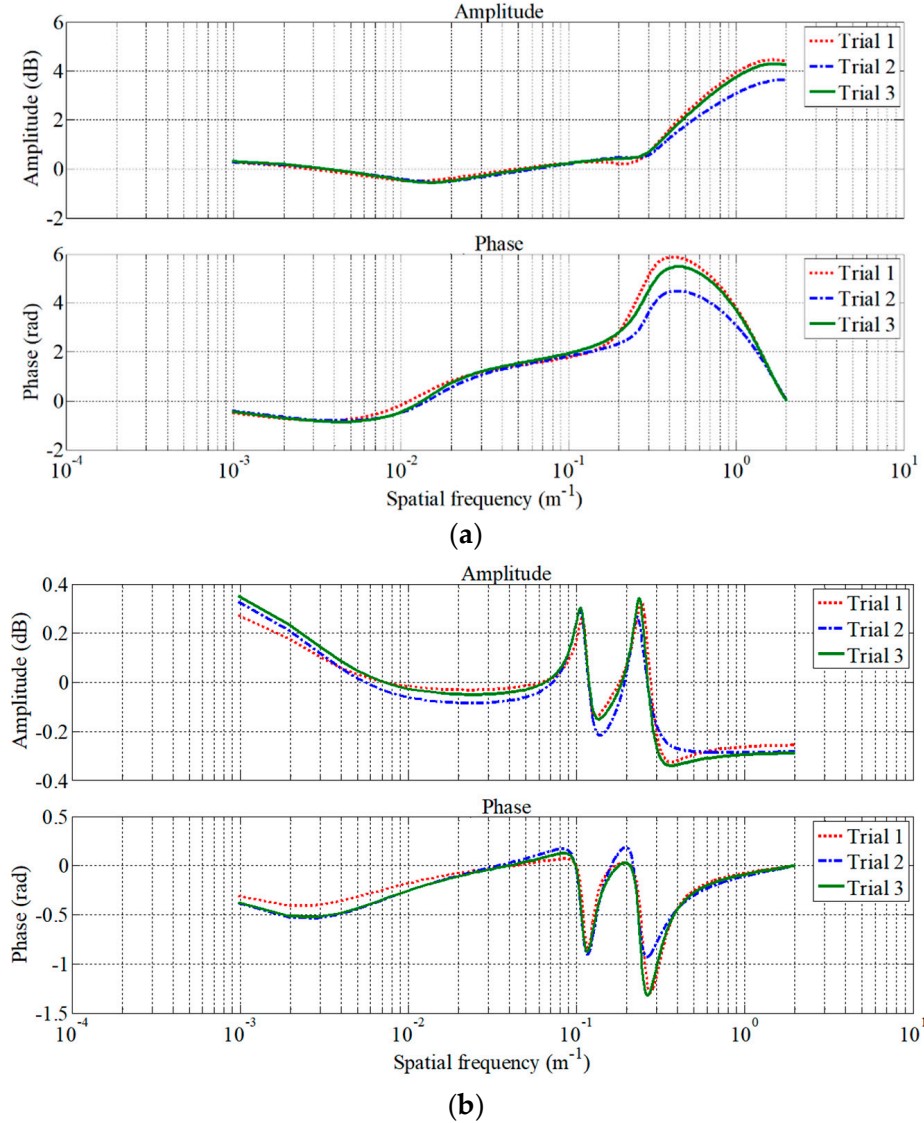

**Figure 7.** Frequency response of the derived models: (**a**) lateral direction; (**b**) vertical direction.

The pole–zero plots of the derived models are shown in Figure 8. All poles are located inside, near the unit circle, and separated from zeros. This ensures that the derived models are stable and can produce bounded signals.

*5.2. Model Validation*

The derived parametric models are applied to estimate track irregularities from the acceleration signals obtained from the HSR-350x train on which the TGIS is installed. The results in the spatial domain are shown in Figure 9. In the lateral direction, the irregularity estimated without the parametric model overestimates the track irregularity, while the derived models ensure the estimated results are close to the measured track irregularity. In the vertical direction, all results are consistent, and no significant improvements are observed regardless of the model used.

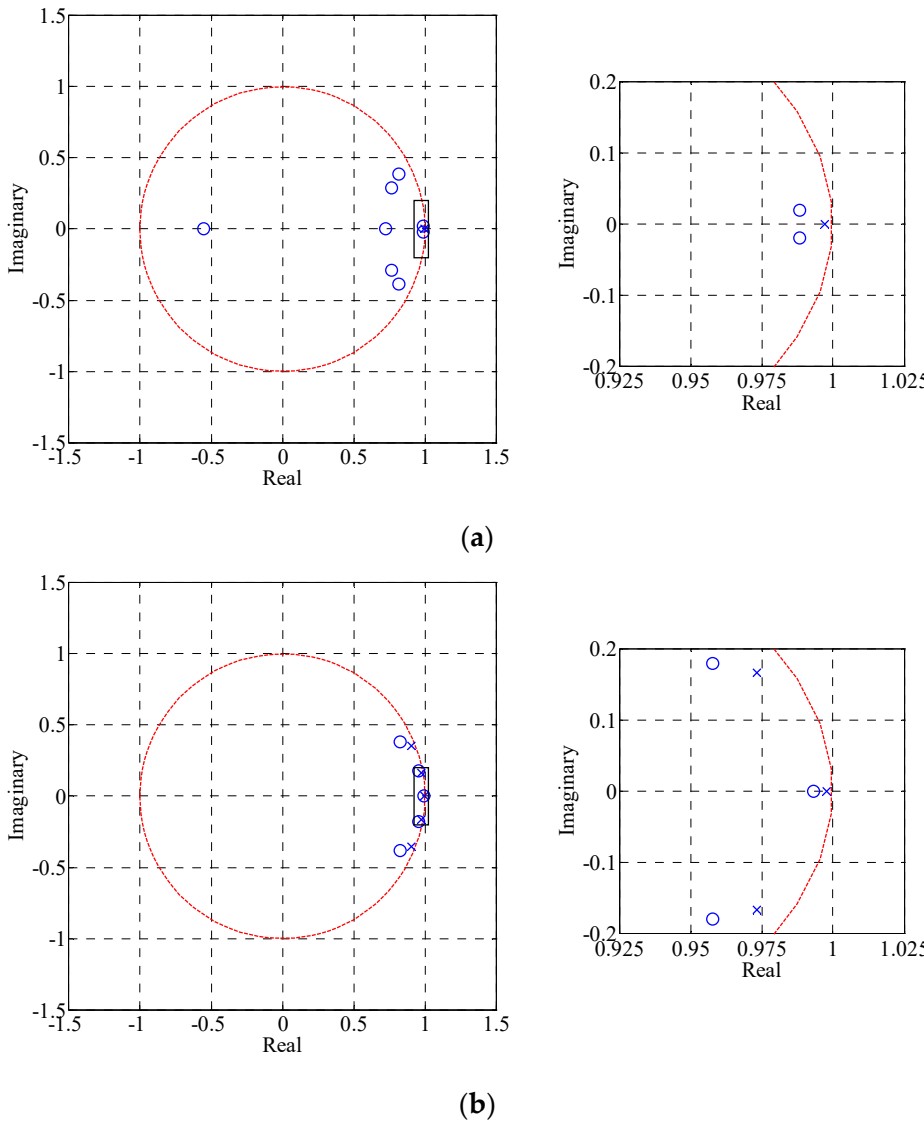

**Figure 8.** Pole−zero plots: (**a**) lateral direction; (**b**) vertical direction.

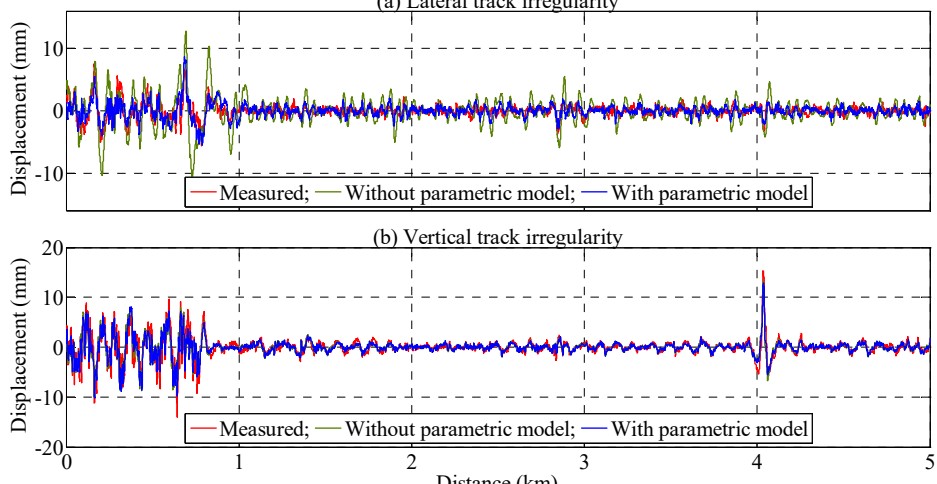

**Figure 9.** Estimation of track irregularities from the TGIS in the spatial domain.

Since it is complicated to analyze the results in the spatial domain, track irregularities are transformed into the wavelength domain, and their power spectral densities are compared as shown in Figure 10. The compensation model improves the irregularity estimation in the lateral direction, especially for wavelengths between 4~70 m. However, the effect of the model is indistinct in the vertical direction.

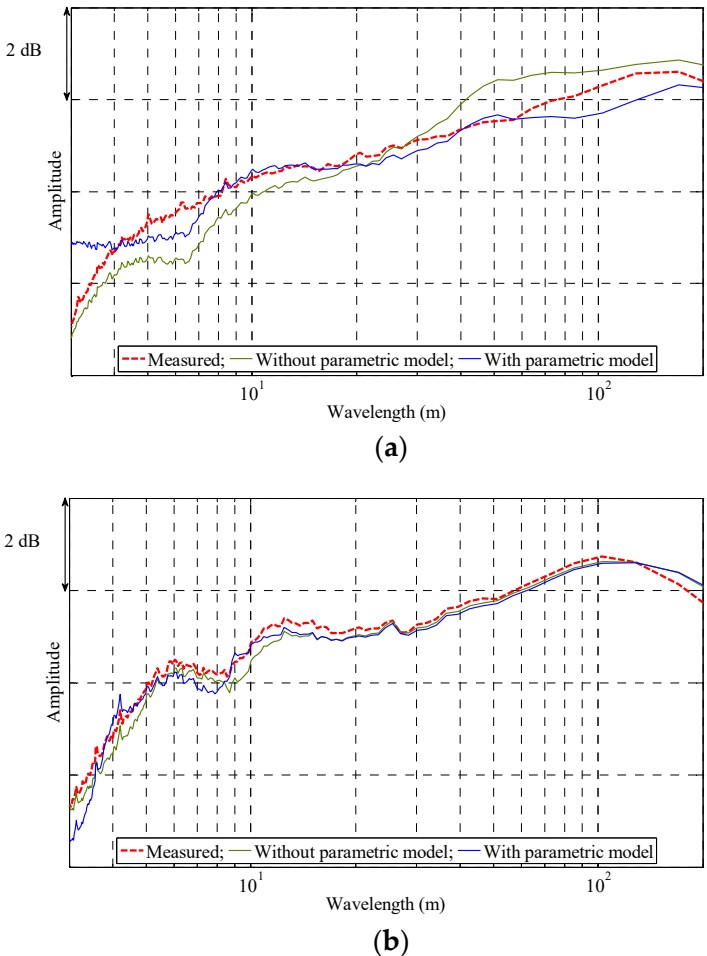

(a)

(b)

**Figure 10.** Estimation of track irregularities using the TGIS signal in the wavelength domain: (**a**) lateral direction; (**b**) vertical direction.

## 6. Track Irregularity Estimation Using Derived Models

To examine the applicability of the derived models, they were used to estimate track irregularities from the signals obtained using the accelerometers installed on the axle box and the bogie of a KTX train in operation. A section of the slab track with notable lateral and vertical track irregularities was selected for comparison. The reference measurement with the TGIS was carried out approximately one year before the measurement tests with the in-service train. The slab track was selected because the variations in time are expected to be smaller than those for the ballasted track. Three measurement tests were carried out within a one-week interval to ensure the reproducibility of the methodology, and the results were compared in the spatial and the wavelength domains as shown in Figures 11–13. The maximum train speed in these test campaigns was 300 km/h. The discrepancies were presumed to be a result of the differences in the suspension characteristics and wheel profiles of the KTX train and HSR-350x train.

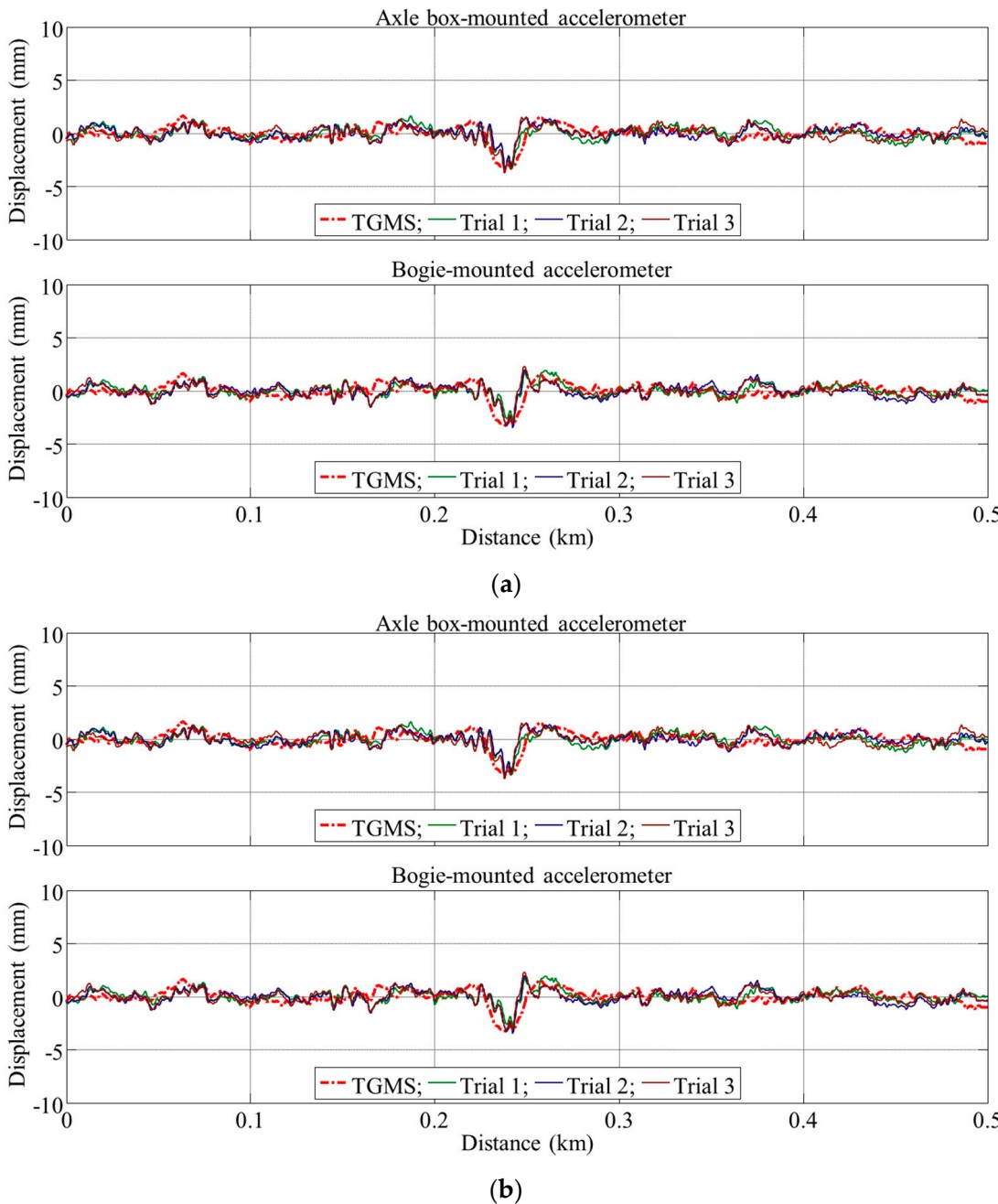

**Figure 11.** Estimation of track irregularities in the spatial domain: (**a**) lateral direction; (**b**) vertical direction.

The estimated and measured lateral track irregularities are compared in Figures 11a and 12. The estimated results from the three tests show excellent agreement, confirming that the proposed method can be used for trains in regular operation. The results also show good agreement with the reference irregularity. In the spatial domain, as shown in Figure 11a, an irregularity is clearly observed near the 0.25 km section from both accelerometers installed on the axle box and the bogie. In the wavelength domain, as shown in Figure 12, the estimated results show good agreement over all wavelengths except below 4 m and near 10 m.

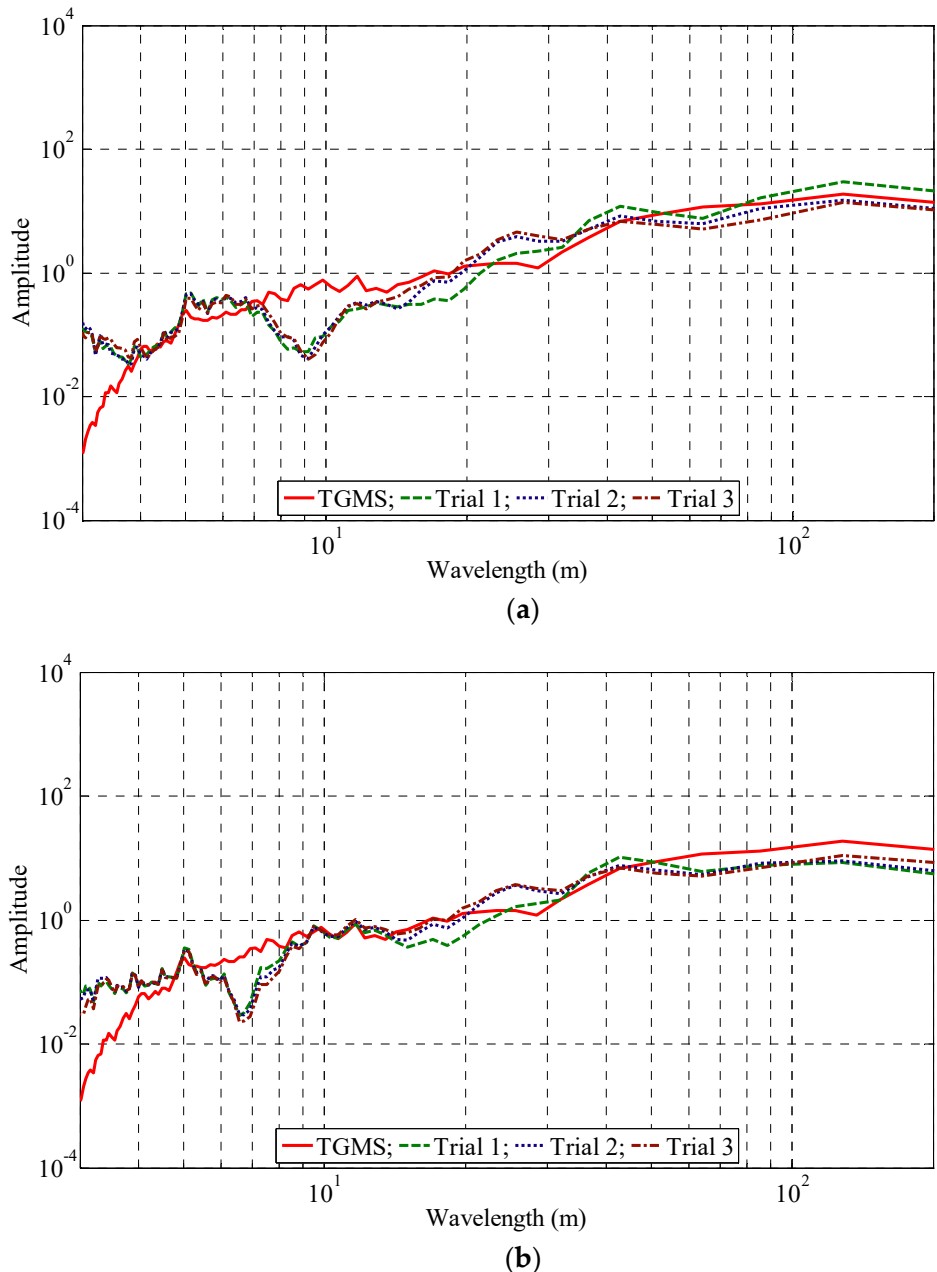

**Figure 12.** Estimation of lateral track irregularities in the wavelength domain: (**a**) axle box-mounted accelerometer; (**b**) bogie-mounted accelerometer.

The estimated and measured vertical track irregularities are compared in Figures 11b and 13. The results from the three tests of the bogie-mounted accelerometer show excellent agreement, while those of the axle box-mounted accelerometer show some discrepancies. However, a notable irregularity near the 0.25 km section is clearly observed. Its magnitude is estimated within a tolerable range in both the axle box and the bogie-mounted accelerometers, as shown in Figure 11b. The wavelength characteristics of the estimated irregularities are shown in Figure 13. The results obtained from the accelerometers installed on the axle box and the bogie exhibit the same spectral characteristics of the measured irregularity. The results show that bogie-mounted accelerometers estimate the irregularities better. It is presumed that the acceleration signal of the axle box is noisier because the vibration level is much higher.

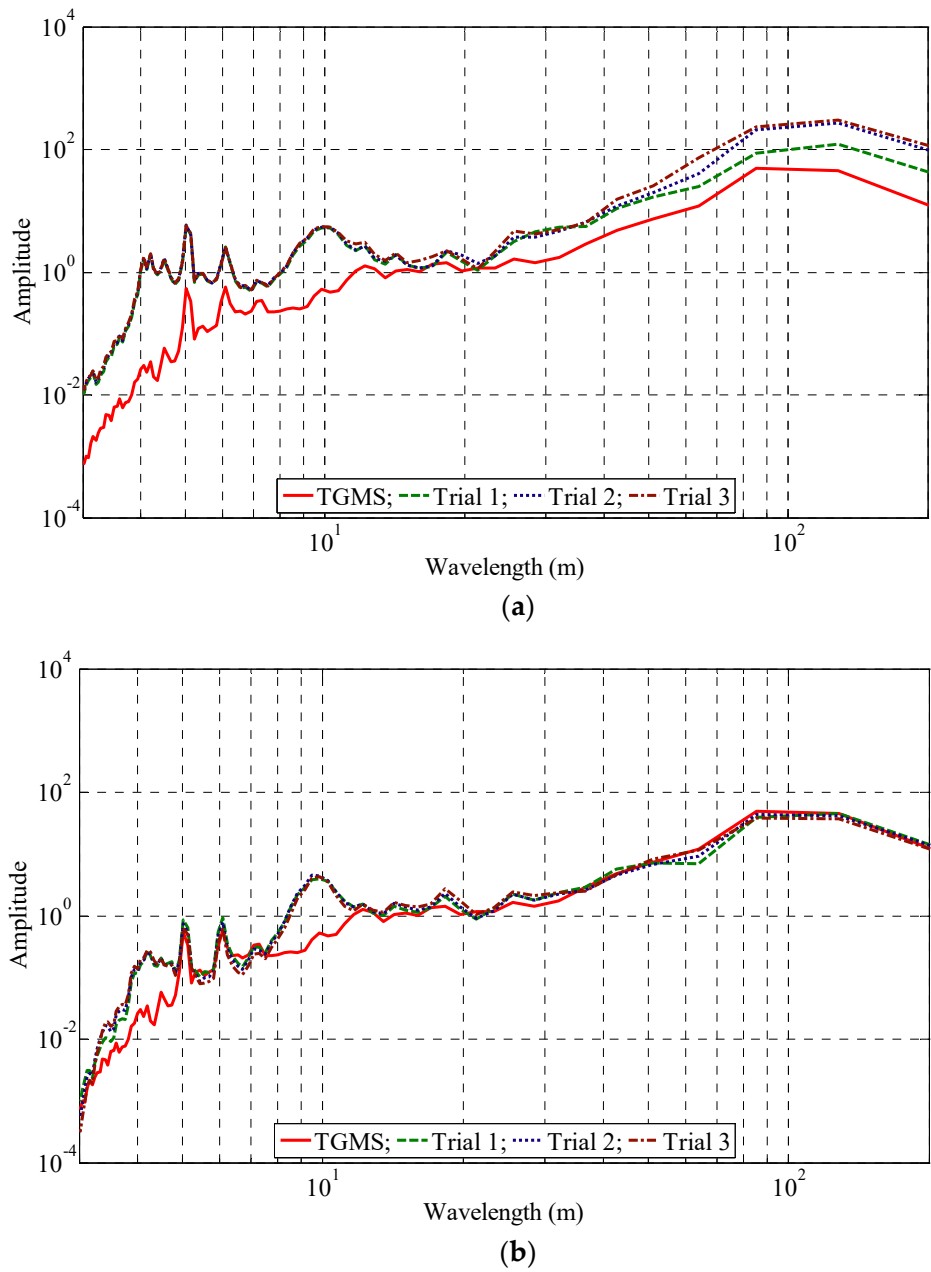

**Figure 13.** Estimation of vertical track irregularities in the wavelength domain: (**a**) axle box-mounted accelerometer; (**b**) bogie-mounted accelerometer.

## 7. Summary and Conclusions

The parametric models are identified by applying a system identification technique that uses estimated displacement from acceleration as the input and measured track irregularity as the output. The parameters are derived from the acceleration and the track irregularities from a track geometry inspection system (TGIS). The parametric models are set up based on the IIR and/or the FIR, and the adaptive Kalman filter is applied for their estimation. The orders of the parametric models are determined by evaluating the PPMCC and the MSE. The number of parameters can be reduced while improving the performance of the models. In this work, a hybrid IIR/FIR model and a single IIR model are selected for lateral and vertical directions. They are validated by estimating irregularities from the acceleration signals measured by the TGIS.

Finally, track irregularities are estimated using acceleration measured from trains in commercial operation. The results using data obtained from three measurement tests show

good agreement, ensuring the methodology's reproducibility. The estimated irregularities are compared with the reference irregularity in the spatial and wavelength domains. The suggested method can detect the location of irregularities in both the lateral and the vertical directions. It is also demonstrated that the estimated irregularities exhibit the same spectral characteristics as the measured irregularity.

In conclusion, the identified parametric models can be used to predict track irregularities from the accelerometers installed on high-speed trains in commercial operation.

**Funding:** This research was supported by a grant from the R&D Program (Development of core technologies for the capsule-type K-500 bogie, PK2203D5) of the Korea Railroad Research Institute.

**Data Availability Statement:** Not applicable.

**Conflicts of Interest:** The authors declare no conflict of interest.

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
