# Peer review of "Identifying Parametric Models Used to Estimate Track Irregularities of a High-Speed Railway"

_machines, doi:10.3390/machines11010006_

Round 1

Reviewer 1 Report

This manuscript presents the efficient parametric models with smaller number of parameters for estimating track irregularities of high-speed railway. The authors initially derive the parameters from the measured acceleration of the train in operation and the track irregularity data obtained by using a specialized track geometry inspection system. Based on the acceleration measured from high-speed rail vehicles in operation, these models are applied to effectively estimate the track irregularities. The effectiveness of the proposed models is validated by the tested results. It is of great significant for the safety operation estimating and the low cost maintenance of high-speed track.

The overall structure of the paper is acceptable, but there are still some deficiencies. They should be considered by the author as follows:

1.      Which type of vehicle was equipped with the ‘Track Geometry Inspection System’? A track inspection vehicle or a high-speed train? This issue should be clearly introduced since the authors claim that “the dynamic characteristics of a special test track inspection vehicle are different from those of high-speed trains in commercial operation, so the dynamic deflection of the track caused by a high-speed train cannot be appropriately measured”. In this manuscript, the parameters of models are derived from the data tested by the TGIS, and used in an in-service high-speed trains, further more validated by TGIS.

2.     In Fig 6(b), the MSE of ‘IIR(6,5) +FIR(4)’ is smaller than that of all the single model, why do the author say ‘a hybrid FIR/IIR model worsens the result’?

3.     What are the factors which can cause change in the Frequency response of the models? And what is the effect of the change on the irregularity estimation? Please add some clarification.

4.     Whether the models are suitable for different track and vehicle type?

5.     What is the train speed considered in deriving the models’ parameters? And whether the parameters be sensitive to train speed? please clarify.

6.     Will the wheel condition (out of roundness and other irregularities) affect the models’ parameters?

7.     From Fig 13, the estimated results overestimate the irregularity, and the author presumed that ‘vertical irregularity, especially the short wavelength components, has progressed during the one-year time interval’, I think it is not reasonable, the models established in the manuscript reflect the relationship between input and output, even though the vertical irregularity has been progressed, the vibration will be changed consequently. That means that the models can not properly estimate the track irregularity when it has been changed significantly? These models need to be revised?

Author Response

Point 1:  Which type of vehicle was equipped with the ‘Track Geometry Inspection System’? A track inspection vehicle or a high-speed train? This issue should be clearly introduced since the authors claim that “the dynamic characteristics of a special test track inspection vehicle are different from those of high-speed trains in commercial operation, so the dynamic deflection of the track caused by a high-speed train cannot be appropriately measured”. In this manuscript, the parameters of models are derived from the data tested by the TGIS, and used in an in-service high-speed trains, further more validated by TGIS.

Response 1: The TGIS was installed on a high-speed train called HSR-350x. HSR-350x was developed as the prototype of commercial high-speed trains (KTX) in Korea. Their dynamic characteristics are not same but similar to each other. This is described in lines 100~102.

Point 2:  In Fig 6(b), the MSE of ‘IIR(6,5) +FIR(4)’ is smaller than that of all the single model, why do the author say ‘a hybrid FIR/IIR model worsens the result’?

Response 2: The model using ‘IIR(6,5) +FIR(4)’ is a hybrid IIR/IFR model which is different from a hybrid FIR/IIR model. The results depend on the order of filters used as described in lines 243~247 and 249~250.

Point 3: What are the factors which can cause change in the Frequency response of the models? And what is the effect of the change on the irregularity estimation? Please add some clarification.

Response 3: In this paper compensation filters (parametric models) are used to compensate for the discrepancies in the amplitude and phase, which are caused by the lateral motion of the axle box or the bogie relative to the track and by the phase delay of the previous filters. Therefore the frequency response in the lateral direction has larger value than that in the vertical direction as shown in Figure 7. This is spelled out in lunes 69-72 and 259~263.

Point 4: Whether the models are suitable for different track and vehicle type?

Response 4: As mentioned in Section 4 (lines 158~160) The relationship between track irregularities and the motion of an axle box or a bogie depends on the dynamic characteristics of the suspension system and the effective wheel conicity. Therefore revision of the model might be required when there are big differences in the properties of the suspension system or the wheel conicity. However the model is independent of the track. In this paper the proposed method was verified on two sites (Figures 9 and 11) while the model parameter was determined on a different track section of 30km length (Figure 4).

Point 5: What is the train speed considered in deriving the models’ parameters? And whether the parameters be sensitive to train speed? please clarify.

Response 5: The acceleration data was measured at the speed of up to 300km/h. The process of obtaining the models’ aprameters are nothing to do with the speed of the train.

Point 6: Will the wheel condition (out of roundness and other irregularities) affect the models’ parameters?

Response 6: This work focus on track irregularities with wavelength larger than 3 m. The circumference length of the wheel is less than 2.7 m (0.86 m diameter) and hence the wheel causes relatively high-frequency short-wavelength acceleration (over 30Hz at 300km/h). Band pass filers are applied to reduce this component (See Figure 2)

Point 7: From Fig 13, the estimated results overestimate the irregularity, and the author presumed that ‘vertical irregularity, especially the short wavelength components, has progressed during the one-year time interval’, I think it is not reasonable, the models established in the manuscript reflect the relationship between input and output, even though the vertical irregularity has been progressed, the vibration will be changed consequently. That means that the models can not properly estimate the track irregularity when it has been changed significantly? These models need to be revised?

Response 7: Your comment is correct. The progress of the irregularity woun’t change the model. Instead discrepancies are presumed to be caused partly by the differences in suspension characteristics and wheel profile of the KTX train and HSR-350x train. It is spelled out in lines 312~314 .

Reviewer 2 Report

The manuscript is interesting and well-written. Research methods are appropriately applied for identifying models for estimating track irregularities of High-speed Railway. The submitted research and its results are an appropriate resource for further research e.g. for railway track with not good maintenance process. It would also be appropriate to comment on the reason for such a large displacement amplitude in Fig. 9.

Author Response

Point 1:  It would also be appropriate to comment on the reason for such a large displacement amplitude in Fig. 9.

Response 1: The estimation using the previous approach (without using parametric model) overestimate the track irregularity, while the estimation using the derived model improves the estimation as described in lines 283~287. It is consistant with the reponse of the parametric shown in Figure 7 which is expained in lines 258~262.

Reviewer 3 Report

please find my attached comments

Author Response

Point 1:  English language problem. There are many mistakes, such as:

1) line 49: to detect? Or to ;

2) line 67: extensive → expensive?

3) line 72: they are used to the acceleration… → the developed models are used in the analysis of acceleration data measured…

Thus, I suggest the author check thoroughly the whole manuscript to remove all these mistakes,  or looking for professional English editing services if needed.

Response 1: Revised as advised. It will be further revised using the editing service provided by MDPI (https://www.mdpi.com/authors/english)

Point 2:  Typos. Two figure 4? Also, in a subplot (a), there are further subplot (a) and (b)? see lines 214-217. I suggest to name these subplots equally and give sufficient descriptions at the caption.

Response 2: Notations (a) and (b) in each subplot are deleted to avoid confusions and the caption has been revised. Figure 7 has been corrected equivalently.

Point 3:  With train speed 320 km/h, sampling frequency 2048 Hz, the spatial sampling interval could be 320/3.6/2048 = 0.043 m. Why it still keeps the sampling interval of 0.25m?

Response 3: I choose to use the same 0.25m which is the spatial resolution of the TGIS. It is spelled out in lines 99-100.

Point 4:  Are the accelerometers only on one side of the axle and bogie?

Response 4: Yes, they are installed on one side.

Point 5:  It looks in Figure 9 and 10 that, the estimations are generally smaller than the measured data, obviously at the peaks, thus underestimate the track irregularities?

Response 5: The estimation in the lateral direction without using parametric model overestimate the track irregularity, while the estimation using the derived model improves the estimation as described in lines 283~287. It is also consistent with the response of the parametric shown in Figure 7 which is explained in lines 258~262.

Point 6:  The track irregularities estimated from axle-box mounted accelerometers and bogie mounted accelerometers are quite different from each other. Which could be more real? Is it really needed using both axle-box mounted accelerometers and bogie mounted accelerometers?

Response 6: Results show that bogie mount accelerometers estimate the irregularities better. I assume one of the reasons is that vibration level as well as the noise level measured on the axle-box are much higher than those measured on the bogie. It is described in lines 331~333.
